# TIGHT CLUSTERS MAKE SPECIALIZED EXPERTS

## ABSTRACT

At the core of Sparse Mixture-of-Experts (MoE) models is the router that learns the clustering structure of the input distribution in order to direct tokens to suitable experts. However these latent clusters may be unidentifiable, causing slow convergence, vulnerability to contamination, and degraded representations. We examine the router through the lens of clustering optimization, deriving optimal feature weights that maximally distinguish these clusters. Using these weights, we compute token-expert assignments in an adaptively transformed space that better separates clusters, helping identify the best-matched expert for each token. In particular, for each expert cluster, we compute weights that scale features according to whether that expert clusters tightly along that feature. We term this novel router the Adaptive Clustering (AC) router. Our AC router confers three connected benefits: 1) faster convergence, 2) better robustness, and 3) overall performance improvement, as experts are specialized in semantically distinct regions of the input space. We empirically demonstrate the advantages of our AC router in language modeling and image classification in both clean and corrupted settings.

## 1 INTRODUCTION

Scaling up model capacity continues to yield performance gains across tasks, notably in visual representation learning and language modeling (Alexey, 2020; Bao et al., 2021; Raffel et al., 2020). However, larger models incur increasing computational cost, prompting research in Sparse Mixture-of-Experts (MoE) models (Shazeer et al., 2017; Fedus et al., 2022; Lepikhin et al., 2020), which activate only sub-modules, or *experts*, to reduce overhead. These models can outperform dense architectures with nearly constant computation on speech recognition, image recognition, machine translation, and language modeling (Riquelme et al., 2021; Kumatani et al., 2021).

At the core of the MoE layer is the learned router which segments the input space such that semantically similar input tokens are assigned to corresponding experts. Recent work explores various routing strategies, from linear programs (Lewis et al., 2021) and cosine similarity-based methods (Chi et al., 2022) to soft assignments (Puigcerver et al., 2023) and top-k routing (Shazeer et al., 2017; Zhou et al., 2022). These methods rely on dot-products between inputs and experts, which can be suboptimal when semantic regions are not easily identified in high-dimensional space. Typically, we expect that the true underlying clusters present in the data will cluster on different, potentially disjoint, subsets of features, and may not be discoverable when using the full feature set. This phenomenon can lead to slower convergence as experts are unable to specialize on semantically similar regions of the data, poor robustness as data contamination can spuriously assign inputs to unsuitable experts, and degraded overall downstream performance due to suboptimal input-expert matching.

**Contribution.** We introduce the Adaptive Clustering (AC) router and Adaptive Clustering Mixture-of-Experts (ACMoE), a novel MoE method in which the router computes token-expert assignments in a transformed space that maximally identifies latent clusters in the data and more easily discovers the best-matched expert for each token. This produces three benefits: 1) *faster convergence* as experts are able to specialize more quickly by being allocated semantically similar inputs, 2) *better robustness* as latent clusters are better separated, thereby minimizing the risk that data corruption erroneously assigns tokens to unsuitable experts, and 3) *better downstream performance* due to improved expert specialization. In order to discover the corresponding weights, we present a feature-weighted clustering optimization perspective on the MoE framework and demonstrate how the clustering solution obtains the required feature weights. We theoretically prove that our proposed routing mechanism learns the latent clustering structure of the data faster than standard routers and that our

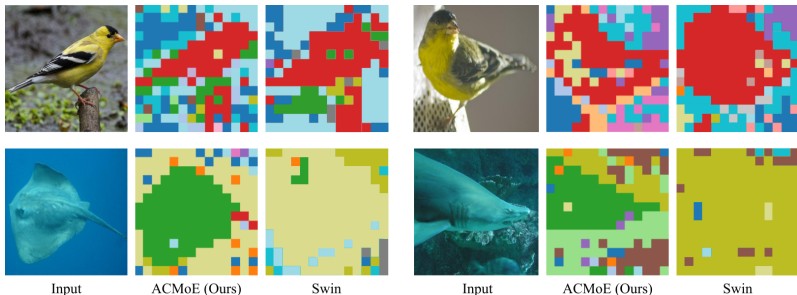

Input    ACMoE (Ours)    Swin    Input    ACMoE (Ours)    Swin

Figure 1: ACMoE discovers semantically distinct regions. We show 14x14 image reconstructions where patches are colored by assigned experts. **Top row:** Swin fails to segment the bird precisely while ACMoE accurately discovers the bird and relevant foreground. **Bottom row:** When the background and foreground are hard to distinguish, Swin fails to register the stingray (left) or shark (right) and allocates one expert for virtually the entire image. ACMoE, however, accurately discovers the semantically distinct regions and utilizes one expert (green) to specialize on the stingray and shark and different experts to specialize on the the background.

mechanism is more robust to data contamination. Furthermore, our proposed method involves no learnable parameters and can be computed highly efficiently. In summary, our contributions are:

1. We develop the novel Adaptive Clustering router for MoE architectures, which computes token-expert assignments in a transformed space that promotes separation of latent clusters in the data and more easily identifies the best-matched expert for each token.

2. We propose a feature-weighted clustering optimization perspective on token-expert assignment and derive the optimal feature weights for routing.

3. We provide a theoretical framework demonstrating how MoE robustness and convergence depend on the clustering structure of the input space.

We empirically demonstrate that 1) the AC router outperforms baseline routers in language modeling and downstream finetuning, and image classification in clean and contaminated settings, 2) the AC router exhibits faster convergence than baseline methods, and 3) the AC router attains these performance improvements with no learnable parameters and negligible computational overhead.

**Preliminaries.** We consider Transformer (Vaswani, 2017) based MoE architectures and follow the approach of previous work where the MoE layer is inserted after the self-attention layer, replacing the traditional feed-forward network (Fedus et al., 2022; Du et al., 2022; Liu et al., 2021). Let $\boldsymbol{h} \in \mathbb{R}^d$ be a hidden representation and $\boldsymbol{e}_1, \boldsymbol{e}_2, \ldots \boldsymbol{e}_N \in \mathbb{R}^d$ be the $N$ learnable expert embeddings for model hidden dimension $d$. The MoE layer selecting the top $k$ experts is described by:

$$\mathcal{K} \coloneqq \mathrm{topk}_k(s_k) = \mathrm{topk}_k(\boldsymbol{h}^\top \boldsymbol{e}_k) \tag{1}$$

$$f^{SMoE}(\boldsymbol{h}) = \boldsymbol{h} + \sum_{k \in \mathcal{K}} g(\boldsymbol{h}^\top \boldsymbol{e}_k) f_k^{\mathrm{FFN}}(\boldsymbol{h}), \tag{2}$$

where $f_k^{\mathrm{FFN}}$ is the $k^{\mathrm{th}}$ expert feed-forward network, $s_k = \boldsymbol{h}^\top \boldsymbol{e}_k$ is the similarity score between token $\boldsymbol{h}$ and the $k^{\mathrm{th}}$ expert $\boldsymbol{e}_k$ and $g(\cdot)$ is a gating function often chosen as softmax. We refer to Eqn. 1 as the router, which learns the best matched experts per token, and Eqn. 2 as the overall MoE layer.

## 2    A CLUSTERING OPTIMIZATION PERSPECTIVE

We analyze the MoE router through the lens of feature-weighted clustering (Witten & Tibshirani, 2010; Friedman & Meulman, 2004; Brusco & Cradit, 2001). We explicitly model the router's task as learning a token assignment that groups together similar tokens. We incorporate learnable feature weights in solving a clustering optimization problem to optimally reveal latent clusters and present an analytical solution for any given routing assignment. We discuss how this solution improves the MoE router before providing the full formulation of our AC router in the next section.

### 2.1    CLUSTERING OPTIMIZATION

Let $\boldsymbol{h}_i = [h_{i1}, \ldots, h_{id}]^\top$ be the $i^{\mathrm{th}}$ hidden representation and $D_{ij}(\boldsymbol{w}) = \sum_{q \in [d]} w_q \rho_{ijq}$ be the distance between pairs of vectors $\boldsymbol{h}_i$ and $\boldsymbol{h}_j$ where $\boldsymbol{w} = [w_1, \ldots, w_d]$ are nonnegative feature weights

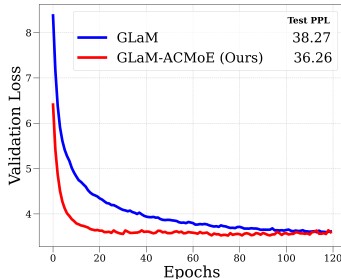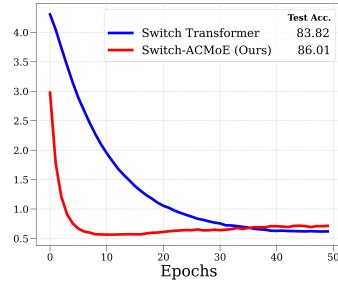

Figure 2: Fast Convergence of ACMoE. **Left:** Convergence speed on WikiText-103 pretraining using GLaM (Du et al., 2022) backbone. **Right:** Convergence speed on Banking-77 finetuning using Switch Transformer (Fedus et al., 2022) backbone. We see faster convergence and better final perplexity (PPL) and accuracy (Acc.).

summing to 1 and $\rho_{ijq}$ is a chosen distance metric over the $q^{th}$ feature. We wish to learn a classifier $r(i) = k$ assigning the $i^{\text{th}}$ object to a group $k$ over the input set of $N$ objects, where objects within the same group are more similar to each other than to those in other groups. We furthermore wish to model the scenario that groupings exist in different latent subspaces with varying dependence on possibly disjoint subsets of features. We therefore use clustering criterion with cluster-dependent feature weights $\{\boldsymbol{w}_k\}_{k=1}^{E}$ for $E$ groups, given by:

$$(r^*, \{\boldsymbol{w}_k^*\}_{k=1}^E) = \arg\min_{r,\{\mathbf{w}_k\}} \sum_{k\in[E]} \frac{1}{N_k^2} \sum_{r(i)=k} \sum_{r(j)=k} D_{ij}^J(\boldsymbol{w}_k),$$

$$\text{such that} \sum_{q\in[d]} w_{qk} = 1, \quad \forall k \in [E], \tag{3}$$

where $D_{ij}^J(\boldsymbol{w}_k) = \sum_{l=1}^d w_{qk}\rho_{ijq} + \lambda J(\boldsymbol{w}_k)$ denotes the weighted distance between $i$ and $j$ combined with regularization $J$ and regularization strength $\lambda$. We set the regularizer to the Kullback-Leibler divergence between $w$ and the uniform distribution $\boldsymbol{u} = (1/d, \ldots, 1/d) \in \mathbb{R}^d$, denoted by $J(\boldsymbol{w}_k) = D_{\text{KL}}(\boldsymbol{u} \parallel \boldsymbol{w}_k)$, where $\lambda$ reflects our preference to maintain more or less features in the solution.

## 2.2 MoE as Clustering Optimization

In the MoE setting, the router performs the role of the classifier $r : \mathbb{R}^d \to [E]$, which is learned via gradient descent on the output loss. Therefore, we fix $r$ and optimize the criterion with respect to cluster-wise feature weights $\boldsymbol{w}_k$. Under this interpretation, the router learns via backpropagation to optimally allocate representations to experts, with representations adaptively transformed to maximally reveal the clustering structure of the input data. The objective in Eqn. 3 then becomes

$$\{\boldsymbol{w}_k^*\}_{k=1}^E = \arg\min_{\{\mathbf{w}_k\}} \sum_{k\in[E]} \frac{1}{N_k^2} \sum_{r(i)=k} \sum_{r(j)=k} D_{ij}^J(\boldsymbol{w}_k), \tag{4}$$

with the same summation to unity constraints as in Eqn. 3. The following theorem presents the optimal weights per feature $q$ and cluster $k$:

**Theorem 1** (Optimal feature weights). *Let $s_{qk} := N_k^{-2} \sum_{r(i)=k} \sum_{r(j)=k} \rho_{ijq}$ be a measure of dispersion on the $q^{\text{th}}$ feature for the representations assigned to cluster $k$. Then, for a given router function $r : \mathbb{R}^d \to [E]$, the corresponding optimal weights $\{\boldsymbol{w}_k\}_{k\in[E]}$ that minimize the feature-weighted clustering optimization problem in Eqn. 4 are given by*

$$w_{qk} = \frac{\lambda/d}{s_{qk} + \alpha_k}, \tag{5}$$

*for* $(q, k) \in [d] \times [E]$, where $\{\alpha_k\}_{k\in[E]}$ are constants that for any $\lambda > 0$ satisfy $\sum_{q\in[d]} \frac{1}{s_{qk}+\alpha_k} = \frac{d}{\lambda}$. The existence $\alpha_k$ and the proof of Theorem 1 is provided in Appendix A.1. The optimal weights for a cluster $k$ given in Eqn. 5 take an intuitive form in that they are inversely proportional to the measure of dispersion in cluster $k$ along each dimension, $\boldsymbol{w}_k \propto \left[\frac{1}{s_{1k}}, \ldots, \frac{1}{s_{dk}}\right]$. Hence, the optimal cluster-wise feature weights scale features according to their contribution to forming tight clusters.

This method enables the MoE router to perform better token-expert matching. The cluster-wise feature weights $\boldsymbol{w}_k$ scale each token according to the specialization of the experts, as large weights indicate those features are highly important to the identification of that expert cluster, thereby allowing the router to best identify the most suitable expert for each token. Note that this solution is local in that we learn the optimal weights adaptively *per cluster*, obtaining $\boldsymbol{w}_k$ for all $k \in [E]$, and so we compute a unique scaling of the feature space adaptively *per cluster* as well.

## 3 A Tight Cluster is a Specialized Expert

In this section, we use the solution from the optimization problem in Eqn. 5 to obtain the AC router and present theoretical results on how AC routing promotes faster convergence and robustness.

### 3.1 Full Technical Formulation

We first integrate the weights from Eqn. 5 into the transformation in Definition 1, which scales each dimension according to the $k^{\text{th}}$ expert's specialization:

**Definition 1** (AC Router Transformation $\boldsymbol{M}_k$). *Let $\mathcal{C}_k^\ell = \{\boldsymbol{h}_1^\ell, \ldots \boldsymbol{h}_{N_k}^\ell\}$ be the tokens assigned to expert $k$ at layer $\ell$. Let $s_{qk}^\ell \in \mathbb{R}$ be a measure of a spread in the $q^{\text{th}}$ dimension for cluster $k$, such as mean absolute deviation $s_{qk}^\ell = \frac{1}{N_k} \sum_{i \in \mathcal{C}_k^\ell} |\boldsymbol{h}_{iq}^\ell - \bar{\boldsymbol{h}}_q^\ell|$. Then, the cluster-dependent router transformation for expert $k$ at layer $\ell$ is given by a diagonal matrix $\boldsymbol{M}_k^\ell := \text{diag}(1/s_{1k}^\ell, \ldots, 1/s_{dk}^\ell)$.*

Using $\boldsymbol{M}_k^\ell$ to adaptively scale the feature space according to the experts' specialization yields our AC router and corresponding ACMoE layer:

**Definition 2** (Adaptive Clustering Router and MoE Layer). *Let $\boldsymbol{h}^\ell \in \mathbb{R}^d$ be the hidden representation of an input, $\boldsymbol{e}_1^\ell, \ldots, \boldsymbol{e}_N^\ell \in \mathbb{R}^d$ be expert embeddings at layer $\ell$. Let $\boldsymbol{h}^{\ell-1} \in \mathcal{C}_{k^*}^{\ell-1}$ have been assigned to expert $k^*$ in the previous layer. Let $\boldsymbol{M}_{k^*}^{\ell-1} \in \mathbb{R}^{d \times d}$ be the Adaptive Clustering transformation (Definition 1) for input $\boldsymbol{h}$ at layer $\ell - 1$. Let $g(\cdot)$ be the softmax function. Then the following equations describe the Adaptive Clustering router (Eqn. 6) and overall ACMoE layer (Eqn. 7):*

$$\mathcal{K} := \text{topk}_k(s_k) = \text{topk}_k(\boldsymbol{h}^{\ell\top} \boldsymbol{M}_{k^*}^{\ell-1} \boldsymbol{e}_k^\ell) \tag{6}$$

$$f^{\text{ACMoE}}(\boldsymbol{h}^\ell) = \boldsymbol{h}^\ell + \sum_{k \in \mathcal{K}} g(\boldsymbol{h}^{\ell\top} \boldsymbol{M}_{k^*}^{\ell-1} \boldsymbol{e}_k^\ell) f_k^{\text{FFN},\ell}(\boldsymbol{h}^\ell). \tag{7}$$

**Remark 1.** *We see from Eqns. 6 and 7 that standard routers and MoE layer are recovered by setting the adaptive clustering router transformation to the identity matrix, $\boldsymbol{M}_k = \boldsymbol{I}_d$ for all $k \in [E]$. Within our framework, standard routers assume all experts $k \in [E]$ depend equally on all dimensions.*

### 3.2 Adaptive Clustering Promotes Robustness and Fast Convergence

We now present theoretical propositions on how our method improves robustness and convergence speed. Robustness follows from the exponentially lower probability of erroneous expert assignment and faster convergence follows from improved Hessian conditioning with respect to expert embeddings. Proofs are deferred to Appendix A.3.

**Robustness.** Lemma 1 shows that our AC transformation (Def. 1) increases inter-cluster separation, and Lemma 2 provides a probability bound for incorrect assignments as a function of inter-cluster distance. Robustness of AC routing then follows as a direct combination of these two lemmas.

**Lemma 1** (Adaptive Clustering Router Transformation Increases Cluster Separation). *Let the data be generated from a Gaussian mixture model with components, $g_c = \mathcal{N}(\boldsymbol{\mu}_c, \boldsymbol{\Sigma}_c)$ for $c \in [E]$. Without loss of generality, consider two expert clusters $c \in \{a, b\}$ where a token representation $\boldsymbol{h} \sim g_a$ belongs to cluster $a$. Let $\boldsymbol{M}_a = \text{diag}(1/s_{1a}, \ldots, 1/s_{da})$ be the router transformation constructed from the feature-wise dispersions, $s_{qa}$, of cluster $g_a$ for each feature $q \in [d]$ as given by Definition 1. Then the distance between cluster means in the $\boldsymbol{M}_a$-transformed space, defined as $\|\boldsymbol{\mu}_k - \boldsymbol{\mu}_a\|_{\boldsymbol{M}_a}^2 := (\boldsymbol{\mu}_k - \boldsymbol{\mu}_a)^\top \boldsymbol{M}_a (\boldsymbol{\mu}_k - \boldsymbol{\mu}_a)$, is larger than in the original Euclidean space: $\|\boldsymbol{\mu}_k - \boldsymbol{\mu}_a\|_{\boldsymbol{M}_a}^2 \geq \|\boldsymbol{\mu}_k - \boldsymbol{\mu}_a\|^2$.*

In Lemma 2, we derive the probability of mis-assignment as a function of inter-cluster distance, highlighting how cluster separation mitigates the effect of noise that can confuse the router.

**Lemma 2** (Incorrect Assignment Probability). *Let $\boldsymbol{h} \sim \mathcal{N}_{k^*}(\boldsymbol{\mu}_{k^*}, \boldsymbol{\Sigma}_{k^*})$ be a representation belonging to cluster $k^*$. Let $\boldsymbol{h}' = \boldsymbol{h} + \boldsymbol{\epsilon}$ be contaminated by some 0-mean noise $\boldsymbol{\epsilon} \sim (\boldsymbol{0}, \boldsymbol{\Sigma}_\epsilon)$. Let $k$ be the nearest, incorrect cluster to $k^*$. Let the inter-cluster mean distance between $k^*$ and $k$ be given by $\|\delta\boldsymbol{\mu}\| := \|\boldsymbol{\mu}_{k^*} - \boldsymbol{\mu}_k\|$. Let the routing assignment be given by $r : \mathbb{R}^d \to [E]$ and denote the cumulative density of a standard normal distribution by $\Phi$. Then the probability of incorrect assignment is*

$$\Pr(r(\boldsymbol{h}') \ne k^*) = 1 - \Phi\left(\frac{\|\delta\boldsymbol{\mu}\|^2}{2\sqrt{\delta\boldsymbol{\mu}^\top(\boldsymbol{\Sigma}_{k^*} + \boldsymbol{\Sigma}_\epsilon)\delta\boldsymbol{\mu}}}\right). \tag{8}$$

It is worth noting that since $1 - \Phi(x) \sim (\sqrt{2\pi}x)^{-1}e^{-x^2/2}$ for large $x$ and $\sqrt{\delta\boldsymbol{\mu}^\top(\boldsymbol{\Sigma}_{k^*} + \boldsymbol{\Sigma}_\epsilon)\delta\boldsymbol{\mu}} = O(\|\boldsymbol{\mu}\|)$, we find that the probability of incorrect cluster assignment as given by Eqn. 8, $\Pr(r(\boldsymbol{h}') \ne k^*) = e^{-O(\|\delta\boldsymbol{\mu}\|^2)}$ is an exponentially decreasing function in $\|\delta\boldsymbol{\mu}\|$. We now combine the notions in Lemmas 1 and 2 to obtain that the probability of erroneous assignment using the AC router is exponentially smaller than under a standard routing scheme:

**Proposition 1** (Robustness of ACMoE). *Consider an expert assignment setting for the representation $\boldsymbol{h} \sim \mathcal{N}_{k^*}(\boldsymbol{\mu}_{k^*}, \boldsymbol{\Sigma}_{k^*})$ as in Lemma 2 with two routers given by $r : \mathbb{R}^d \to [E]$ and $r^{\mathrm{AC}} : \mathbb{R}^d \to [E]$ for standard (Eqn. 2) and AC routers (Definition 2), respectively. Then the probabilities of incorrect assignments of routers $r$ and $r^{\mathrm{AC}}$ satisfy $\Pr\left(r^{\mathrm{AC}}(\boldsymbol{h}') \ne k^*\right) \le \Pr\left(r(\boldsymbol{h}') \ne k^*\right)$.*

**Convergence.** Our AC router reduces the conditioning number of the Hessian of the loss with respect to the expert $\boldsymbol{e}_k$, improving the loss landscape and enabling faster convergence of the router. We find this result empirically supported, as shown in Fig. 2. Formally this is:

**Proposition 2** (Faster convergence of ACMoE). *Let $\mathcal{L}^{\mathrm{MoE}} : \boldsymbol{\Theta} \to \mathbb{R}_+$ and $\mathcal{L}^{\mathrm{ACMoE}} : \boldsymbol{\Theta} \to \mathbb{R}_+$ be the network loss functions over parameters $\boldsymbol{\Theta}$ when employing the standard (Eqn. 2) and AC routers (Definition 2), respectively. Let $\kappa(\boldsymbol{A}) = \lambda_{\max}/\lambda_{\min}$ denote the conditioning number of a matrix $\boldsymbol{A}$ with largest and smallest eigenvalues $\lambda_{\max}$ and $\lambda_{\min}$ respectively. Let the Hessian of an $i^{\mathrm{th}}$ expert be given by $\nabla^2_{\boldsymbol{e}_i}$. Then for each $i \in [E]$ the following holds with high probability*

$$\kappa\left(\nabla^2_{\boldsymbol{e}_i}\mathcal{L}^{\mathrm{ACMoE}}\right) \le \kappa\left(\nabla^2_{\boldsymbol{e}_i}\mathcal{L}^{\mathrm{MoE}}\right) \tag{9}$$

# 4 EXPERIMENTAL RESULTS

We evaluate our method on Wikitext-103 (Merity et al., 2016) language modeling and ImageNet (Deng et al., 2009) image classification. We integrate our AC router into Switch Transformer (Fedus et al., 2022), Generalist Language Model (GLaM) (Du et al., 2022), and Swin Transformer (Liu et al., 2021) backbones. We show i) ACMoE obtains substantive improvements over baseline models across both language and vision tasks; ii) ACMoE offers robust improvements on contaminated samples. We additionally show in Appendix B that ACMoE attains these gains with negligible additional computational overhead. Results are averaged over 5 runs with different seeds.

## 4.1 LANGUAGE MODELING

**Setup.** Following Pham et al. (2024), we compare ACMoE against Switch Transformer and GLaM using 16 experts and top-2 routing with 220M parameters. We report pretraining test perplexity (PPL) for Wikitext-103 and top-1 accuracy for finetuning classification tasks on Stanford Sentiment Treebank-2 (SST2) (Socher et al., 2013), Stanford Sentiment Treebank-5 (SST5) (Socher et al., 2013), and Banking-77 (B77) (Casanueva et al., 2020). Full details are provided in Appendix C.

**Language Modeling Results.** Table 2 show test PPL on clean WikiText-103 and when contaminated by Text Attack, where words are randomly swapped with a token 'AAA'. We follow the setup of Han et al. (2024) and assess models by training them on clean data before attacking the test data. ACMoE outperforms baseline Switch and GlaM in both clean and contaminated settings with gains of up to 5.8%. Table 1 further shows ACMoE pretrained models surpass the performance of baselines in finetuning, with strong, consistent improvements of approximately 3%.

## 4.2 IMAGE CLASSIFICATION

**Setup.** Following Liu et al. (2021), we evaluate ACMoE against a 280M parameter Swin Transformer with 16 experts. We evaluate robustness under white box attacks fast gradient sign method

Table 1: WikiText-103 test PPL and top-1 test accuracy on SST2, SST5, and B77 finetuning.

| Model | Test PPL (↓) | SST2 (↑) | SST5 (↑) | B77 (↑) |
|---|---|---|---|---|
| *Switch Transformer* (Fedus et al., 2022) | 35.48 | 76.27 | 39.13 | 83.82 |
| Switch-ACMoE (**Ours**) | **34.42** | **77.32** | **40.04** | **86.01** |
| *GLaM* (Du et al., 2022) | 38.27 | 69.97 | 33.69 | 80.89 |
| GLaM-ACMoE (**Ours**) | **36.26** | **71.90** | **34.24** | **82.33** |

Table 2: Clean and Contaminated Test Perplexity (PPL) on WikiText-103

| Model | Clean Test PPL (↓) | Contaminated Test PPL (↓) |
|---|---|---|
| *Switch Transformer* (Fedus et al., 2022) | 35.48 | 48.12 |
| Switch-ACMoE (**Ours**) | **34.42** | **47.61** |
| *GLaM* (Du et al., 2022) | 38.27 | 50.84 |
| GLaM-ACMoE (**Ours**) | **36.26** | **47.91** |

Table 3: Test Accuracy on ImageNet corrupted PGD, FGSM, and SPSA.

| Model | Clean Data | | PGD | | FGSM | | SPSA | |
|---|---|---|---|---|---|---|---|---|
| | Top 1 | Top 5 | Top 1 | Top 5 | Top 1 | Top 5 | Top 1 | Top 5 |
| *Swin* (Liu et al., 2021) | 76.10 | 92.99 | 40.85 | 75.51 | 54.70 | 85.22 | 60.57 | 82.75 |
| Swin-ACMoE (**Ours**) | **76.31** | **93.14** | **43.74** | **78.55** | **55.78** | **85.80** | **63.47** | **86.05** |

(FGSM) (Goodfellow et al., 2014) and projected gradient descent (PGD) (Madry et al., 2017), and black box simultaneous perturbation stochastic approximation (SPSA) (Uesato et al., 2018).

**Image Classification Results..** Table 3 shows performance on ImageNet against FGSM, PGD, and SPSA. Compared against Swin Transformer, ACMoE improves a noteworthy 7% against PGD.

## 5 RELATED WORK

**Routing Methods.** Recent studies propose routers based on reinforcement learning (Bengio et al., 2015), linear programs (Lewis et al., 2021; Nguyen et al., 2024), cosine similarity (Chi et al., 2022), greedy top-k experts per token (Shazeer et al., 2017) and greedy top-k tokens per expert (Zhou et al., 2022). These works have predominantly considered dot-products as a suitable similarity metric. This work continues with dot-product based learnable routing but computes the routing assignments in an adaptively transformed space to maximally identify the latent expert clusters.

**MoE and Cluster Analysis.** Recent studies on MoE show the router can recover the clustering structure of the input space and each expert specializes in a specific cluster (Dikkala et al., 2023; Chen et al., 2022). Our work considers transformations of the input space to identify expert clusters, and we learn these transformations via feature-weighted cluster analysis (Brusco & Cradit, 2001; Witten & Tibshirani, 2010; Gnanadesikan et al., 1995). Friedman & Meulman (2004) consider cluster-dependent feature weights to augment iterative clustering algorithms. Our approach similarly uses cluster-dependent feature weights but uses a different optimization problem to derive optimal weights that directly capture the importance of each feature to the clustering solution.

## 6 CONCLUSION AND FUTURE WORK

In this paper, we present the Adaptive Clustering (AC) router and ACMoE layer, a novel MoE routing method that computes token-expert assignments in a transformed space that maximally identifies latent clusters in the data and more easily discovers the best-matched expert for each token. We adaptively learn for each input which features are relevant to determining its latent cluster assignment and scale its features accordingly, where features that promote tight clustering are upweighted. Our AC routing method enables faster convergence by improving the Hessian conditioning of the router and better robustness by increasing the separation of latent clusters in the transformed space. For ongoing work, we are investigating improved methods for estimating the latent cluster memberships without reliance on previous layers and with provable consistency guarantees.

**Reproducibility Statement.** Source code for our experiments are provided in the supplementary material. We provide the full details of our experimental setup – including datasets, model specification, train regime, and evaluation protocol – for all experiments in Appendix C. All datasets are publicly available.

**Ethics Statement.** Our work considers fundamental architectures, and in particular their robustness and convergence properties. Given this, we foresee no issues regarding fairness, privacy, or security, or any other harmful societal or ethical implications in general.

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

# Supplement to "Tight Clusters Make Specialized Experts"

**Table of Contents**

## A   Technical Proofs

### A.1   Proof of Theorem 1

To begin with, we present the following lemma to show the existence of constants $\alpha_k$ for $k \in [E]$ that satisfy Eqn. **??**:

**Lemma 3.** *For any $\lambda > 0$, Eqn.* **??** *has exactly $d$ real solutions with respect to $\alpha_k$.*

*Proof of Lemma 3.* Without loss of generality, assume that $s_{1k} \geq s_{2k} \geq \cdots \geq s_{dk}$. Denote

$$\varphi(\alpha) := \sum_{q \in [d]} \frac{1}{s_{qk} + \alpha} - \frac{d}{\lambda}. \tag{10}$$

Then, the existence of solutions to Eqn. **??** is equivalent to the condition $\varphi(\alpha_l) = 0$. Note that $\varphi(\alpha)$ is a strictly decreasing function in its connected continuity domains since

$$\varphi'(\alpha) = -\sum_{q \in [d]} \frac{1}{(s_{qk} + \alpha)^2} < 0 \tag{11}$$

for all $\alpha \in \mathbb{R} \setminus \{-s_{1k}, \ldots, -s_{dk}\}$. Further, we observe that

$$\lim_{\alpha \to -s_{qk}^-} \varphi(\alpha) = -\infty, \qquad \lim_{\alpha \to -s_{qk}^+} \varphi(\alpha) = +\infty \tag{12}$$

for all $q \in [d]$, and

$$\lim_{\alpha \to \pm\infty} \varphi(\alpha) = -\frac{d}{\lambda} < 0. \tag{13}$$

Now consider the domain of continuity of $\varphi(\alpha)$, namely $(-\infty, -s_{1k}) \cup (-s_{1k}, -s_{2k}) \cup \cdots \cup (-s_{dk}, \infty)$. Due to the monotonicity and limits 12 & 13, there exists a unique solution in each of the intervals except for $(-\infty, -s_{1k})$ where the function is always strictly negative, thus, yielding $d$ roots in total.
□

Now we follow up with the main proof of this section.

*Proof of Theorem 1.* First, let $\mathcal{I}_k := \{i : r(i) = k\}$ for convenience. Now let us restate the clustering optimization problem (3) here once again:

$$\min_{\boldsymbol{w}_k} Q(c, \{\boldsymbol{w}_k\}_{k \in [E]}) = \sum_{k \in [E]} \frac{1}{N_k^2} \sum_{i,j \in \mathcal{I}_k} \sum_{q \in [d]} \left( w_{qk} \rho_{ijq} + \frac{\lambda}{d} \log \frac{1}{dw_{qk}} \right),$$

$$\text{such that} \sum_{q \in [d]} w_{qk} = 1, \quad \forall k \in [E], \tag{14}$$

where we have immediately used the fact that

$$D_{\text{KL}}(\boldsymbol{u} \,\|\, \boldsymbol{w}_k) = \sum_{q \in [d]} \frac{1}{d} \log \frac{1/d}{w_{qk}}. \tag{15}$$

Also, note that

$$\sum_{q \in [d]} \left( w_{qk} \rho_{ijq} + \lambda \frac{1}{d} \log \frac{1}{dw_{qk}} \right) = \sum_{q \in [d]} \left( w_{qk} \rho_{ijq} - \lambda \frac{1}{d} \log(dw_{qk}) \right)$$

$$= \sum_{q \in [d]} \left( w_{qk} \rho_{ijq} - \frac{\lambda}{d} \log w_{qk} \right) - \lambda \log d. \tag{16}$$

We can ignore the term $\lambda \log d$ since it does not depend on the optimization variable. Method of Lagrange multipliers turns this constrained optimization problem into the following unconstrained counterpart:

$$\min_{\boldsymbol{w}_k, \boldsymbol{\alpha}} \mathcal{L}(c, \{\boldsymbol{w}_k\}_{k \in [E]}, \boldsymbol{\alpha}) = \sum_{k \in [E]} \frac{1}{N_k^2} \sum_{i,j \in \mathcal{I}_k} \sum_{q \in [d]} \left( w_{qk} \rho_{ijq} - \frac{\lambda}{d} \log w_{qk} \right) + \sum_{k \in [E]} \alpha_k \left( \sum_{q \in [d]} w_{qk} - 1 \right),$$

where $\boldsymbol{\alpha} = \begin{bmatrix} \alpha_1 & \cdots & \alpha_L \end{bmatrix}^\top$ is the vector of Lagrange multipliers. Note that the last optimization problem can be separated into the following $L$ independent optimization subproblems:

$$\min_{\boldsymbol{w}_k, \boldsymbol{\alpha}} \mathcal{L}_k(c, \boldsymbol{w}_k, \boldsymbol{\alpha}) = \frac{1}{N_k^2} \sum_{i,j \in \mathcal{I}_k} \sum_{q \in [d]} \left( w_{qk} \rho_{ijq} - \frac{\lambda}{d} \log w_{qk} \right) + \alpha_k \left( \sum_{q \in [d]} w_{qk} - 1 \right),$$

for $k \in [E]$. Since the objective function is a positive combination of convex functions, the optimization problem is also convex. By setting the derivatives of $\mathcal{L}_k$ with respect to both optimization variables to 0, we obtain the following system of equations:

$$\begin{cases} \dfrac{\partial \mathcal{L}_k}{\partial w_{qk}} = s_{qk} - \dfrac{\lambda}{d} \dfrac{1}{w_{qk}} + \alpha_k = 0, \\ \dfrac{\partial \mathcal{L}_k}{\partial \alpha_k} = \sum_{q\in[d]} w_{qk} - 1 = 0 \end{cases}$$

for all $k \in [E]$, where $s_{qk}$ is the data dispersion measure defined in the theorem statement. The first equation yields

$$w_{qk} = \frac{\lambda}{d} \frac{1}{s_{qk} + \alpha_k}, \tag{17}$$

where $\alpha_k$ is found from $\sum_{q\in[d]} w_{qk} = 1$ which in fact gives

$$\sum_{q\in[d]} \frac{1}{s_{qk} + \alpha_k} = \frac{d}{\lambda} \tag{18}$$

for all $k \in [E]$ as desired. $\qquad\square$

## A.2 PROOF OF PROPOSITION 1

To give the probability bound an exact form, we assume the clusters follow a Gaussian mixture model (GMM). We note that GMMs are a highly expressive and general framework, so this assumption does not place significant restrictions on our analysis. We further assume that though clusters may overlap, they are well-separated along the features for which they cluster tightly[1].

Since Proposition 1 is a composition of Lemma 1 and Lemma 2, we proceed by providing their proofs.

### A.2.1 PROOF OF LEMMA 1

*Proof of Lemma 1.* Notice that we can expand inequality (1) as

$$\sum_{i\in[d]} m_i \delta\mu_i^2 \geq \sum_{i\in[d]} \delta\mu_i^2,$$

where we let $\delta\boldsymbol{\mu} := \boldsymbol{\mu}_b - \boldsymbol{\mu}_a$. Since $\boldsymbol{M}_a$ entries are mean-scaled, we can rewrite them as

$$m_i = \frac{dm_i'}{\sum_{j\in[d]} m_j'} \tag{19}$$

for some initial dispersion estimates $\{m_j'\}_{j\in[d]}$. Without loss of generality, assume that $[d']$ is the set of dimension indices for which the dispersions are relatively much smaller than those in the rest of the dimensions in the sense that $m_i' \gg m_j'$ for any $i \in [d']$ and $j \in [d] \setminus [d']$. Then, there exists a positive $\alpha \ll 1/2$ such that $\sum_{i\in[d']} m_i > d - \alpha$ and $\sum_{i\in[d]\setminus[d']} m_i < \alpha$. By the assumption that clusters are best-separated along the features for which they cluster tightly, this means that the weight matrix $\boldsymbol{M}_a$ maximizes the contribution of largest $d'$ terms in $\sum_{i\in[d]} m_i \delta\mu_i^2$ corresponding to individual feature-wise distances in dimensions where the feature dispersions are the smallest instead of giving uniform weights to all dimensions, which leads to inequality (1). $\qquad\square$

### A.2.2 PROOF OF LEMMA 2

*Proof of Lemma 2.* Since we use the $\mathcal{L}_2$ distance between the token $\boldsymbol{h}$ and $\boldsymbol{\mu}_c$ as a similarity metric, we assign cluster $g_{k^*}$ to the token $\boldsymbol{h}'$ iff $\|\boldsymbol{h}' - \boldsymbol{\mu}_{k^*}\| \leq \|\boldsymbol{h}' - \boldsymbol{\mu}_k\|$. Assume that the token $\boldsymbol{h}'$ is a noisy observation of an underlying true token $\boldsymbol{h}$ which actually originates from cluster $g_{k^*}$. Then, the token $\boldsymbol{h}'$ can be decomposed as $\boldsymbol{h}' = \boldsymbol{h} + \boldsymbol{\epsilon}$ for a random noise $\boldsymbol{\epsilon} \sim \mathcal{N}(\boldsymbol{0}, \boldsymbol{\Sigma}_\epsilon)$. Now define

---

[1]Intuitively, this assumption captures the natural property that the semantic regions of the input space are distinct along the dimensions that best identify them.

the decision variable $\mathcal{D}(\boldsymbol{h}') := \|\boldsymbol{h}' - \boldsymbol{\mu}_{k^*}\|^2 - \|\boldsymbol{h}' - \boldsymbol{\mu}_k\|^2$ which turns the clustering condition to $\mathcal{D}(\boldsymbol{h}') \le 0$ for the cluster $g_{k^*}$. Let us analyze the decision variable $\mathcal{D}$ as a random variable where randomness may come from the underlying sampling strategy and noise. Note that

$$
\begin{aligned}
\mathcal{D}(\boldsymbol{h}') &= \|\boldsymbol{h} + \boldsymbol{\epsilon} - \boldsymbol{\mu}_{k^*}\|^2 - \|\boldsymbol{h} + \boldsymbol{\epsilon} - \boldsymbol{\mu}_k\|^2 \\
&= \|\boldsymbol{h} - \boldsymbol{\mu}_{k^*}\|^2 - \|\boldsymbol{h} - \boldsymbol{\mu}_k\|^2 + 2(\boldsymbol{\mu}_k - \boldsymbol{\mu}_{k^*})^\top \boldsymbol{\epsilon} \\
&= \mathcal{D}(\boldsymbol{h}) + 2\delta\boldsymbol{\mu}^\top \boldsymbol{\epsilon},
\end{aligned}
\tag{20}
$$

where $\delta\boldsymbol{\mu} := \boldsymbol{\mu}_k - \boldsymbol{\mu}_{k^*}$. Due to the assumption that $\boldsymbol{h}$ is drawn from the distribution $g_{k^*}$, it can be rewritten as $\boldsymbol{h} = \boldsymbol{\mu}_{k^*} + \boldsymbol{\nu}$ with $\boldsymbol{\nu} \sim \mathcal{N}(\boldsymbol{0}, \boldsymbol{\Sigma}_{k^*})$. Then for the first term in Eqn. 20, we have

$$
\begin{aligned}
\mathcal{D}(\boldsymbol{h}) &= \|\boldsymbol{h} - \boldsymbol{\mu}_{k^*}\|^2 - \|\boldsymbol{h} - \boldsymbol{\mu}_k\|^2 \\
&= \delta\boldsymbol{\mu}^\top (2\boldsymbol{h} - \boldsymbol{\mu}_{k^*} - \boldsymbol{\mu}_k) \\
&= \delta\boldsymbol{\mu}^\top (2\boldsymbol{\nu} - \delta\boldsymbol{\mu}) \\
&= 2\delta\boldsymbol{\mu}^\top \boldsymbol{\nu} - \|\delta\boldsymbol{\mu}\|^2.
\end{aligned}
\tag{21}
$$

Substituting this back into Eqn. 20, we get

$$
\mathcal{D}(\boldsymbol{h}') = 2\delta\boldsymbol{\mu}^\top (\boldsymbol{\nu} + \boldsymbol{\epsilon}) - \|\delta\boldsymbol{\mu}\|^2.
\tag{22}
$$

This shows that $\mathcal{D}(\boldsymbol{h}') \sim \mathcal{N}\left(-\|\delta\boldsymbol{\mu}\|^2, 4\delta\boldsymbol{\mu}^\top (\boldsymbol{\Sigma}_{k^*} + \boldsymbol{\Sigma}_\epsilon)\delta\boldsymbol{\mu}\right)$. Since $\mathcal{D}(\boldsymbol{h}')$ follows a normal distribution with the derived parameters, the probability that $\boldsymbol{h}'$ is assigned to cluster $g_{k^*}$ is given by

$$
\Pr(\text{correct cluster}) = \Pr\left(\mathcal{D}(\boldsymbol{h}) \le 0\right) = \Phi\left(\frac{\|\delta\boldsymbol{\mu}\|^2}{2\sqrt{\delta\boldsymbol{\mu}^\top (\boldsymbol{\Sigma}_{k^*} + \boldsymbol{\Sigma}_\epsilon)\delta\boldsymbol{\mu}}}\right),
\tag{23}
$$

where $\Phi$ denotes the CDF of normal distribution as usual. Since $\Phi$ is an increasing function, the probability that the noisy token $\boldsymbol{h}$ is assigned to the correct cluster is proportional to the distance between the cluster centroids and inverse proportional to the covariance matrices of the cluster and the additive noise. On the other hand, for the incorrect clustering probability, we have

$$
\Pr(\text{incorrect cluster}) = 1 - \Phi\left(\frac{\|\delta\boldsymbol{\mu}\|^2}{2\sqrt{\delta\boldsymbol{\mu}^\top (\boldsymbol{\Sigma}_{k^*} + \boldsymbol{\Sigma}_\epsilon)\delta\boldsymbol{\mu}}}\right)
\tag{24}
$$

as claimed. $\qquad\square$

### A.3 Proof of Proposition 2

*Proof of Proposition 2.* Let the router be given by $g$ and let the softmax function be given by $g_{\boldsymbol{\theta}} : \mathbb{R}^d \to \mathbb{R}^d$, parameterized by expert embeddings $\{\boldsymbol{e}_i\}_{i \in [E]}$. The network loss depends on expert embeddings only through the router function $g$. We shall explore the exclusive contribution of each expert embedding in minimizing $\mathcal{L}^{\text{ACMoE}}$. In order to do this, we look at the network loss as a scalar function of $i^{\text{th}}$ expert embedding vector while treating all other network parameters as fixed. Then, we can write $\mathcal{L}^{\text{ACMoE}} : \mathbb{R}^d \to \mathbb{R}$ such that $\mathcal{L}^{\text{ACMoE}} = \mathcal{L}^{\text{ACMoE}}(g_{\boldsymbol{\theta}}(\boldsymbol{e}_i))$. For simplicity, we shall omit the subscript $\boldsymbol{\theta}$. The gradient that comes from back-propagation is then given by

$$
\nabla_{\boldsymbol{e}_i} \mathcal{L}^{\text{ACMoE}} = \left(\nabla_g \mathcal{L}^{\text{ACMoE}}\right)^\top \nabla_{\boldsymbol{e}_i} g,
\tag{25}
$$

where $\nabla_{\boldsymbol{e}_i} g \in \mathbb{R}^{d \times d}$ denotes the Jacobian matrix of $g$ since for $g_k := (g_{\boldsymbol{\theta}}(\boldsymbol{e}_i))_k$, we can write

$$
\frac{\partial}{\partial e_{is}} \mathcal{L}^{\text{ACMoE}}(g_1, \dots, g_d) = \sum_k \frac{\partial \mathcal{L}^{\text{ACMoE}}}{\partial g_k} \frac{\partial g_k}{\partial e_{is}}.
\tag{26}
$$

Note that for $g_k = \text{softmax}(\boldsymbol{h}^\top \boldsymbol{M} \boldsymbol{e}_k)$, we have

$$
\frac{\partial g_k}{\partial e_{is}} = m_s h_s g_k (\delta_{ki} - g_i) = m_s h_s b_{ki}.
\tag{27}
$$

Then, the element of the Hessian matrix of the network loss at index $(s,t) \in [d] \times [d]$ can be written as

$$
\begin{aligned}
\boldsymbol{H}_{st}^{(i)}(\mathcal{L}^{\text{ACMoE}}) = \frac{\partial^2 \mathcal{L}^{\text{ACMoE}}}{\partial e_{is} \partial e_{it}} &= \frac{\partial}{\partial e_{it}} \sum_k \frac{\partial \mathcal{L}^{\text{ACMoE}}}{\partial g_k} \frac{\partial g_k}{\partial e_{is}} \\
&= \sum_k \left( \sum_j \frac{\partial^2 \mathcal{L}^{\text{ACMoE}}}{\partial g_k \partial g_j} \frac{\partial g_j}{\partial e_{it}} \right) \frac{\partial g_k}{\partial e_{is}} + \frac{\partial \mathcal{L}^{\text{ACMoE}}}{\partial g_k} \frac{\partial^2 g_k}{\partial e_{is} \partial e_{it}} \\
&= m_s h_s m_t h_t \left[ \sum_k \left( \sum_j \frac{\partial^2 \mathcal{L}^{\text{ACMoE}}}{\partial g_k \partial g_j} b_{ji} \right) b_{ki} + \frac{\partial \mathcal{L}^{\text{ACMoE}}}{\partial g_k} b'_{ki} \right] \\
&= m_s h_s m_t h_t B_i,
\end{aligned} \tag{28}
$$

where $B_i$ is some constant that depends only on index $i$. Due to Eqn. 28, the Hessian takes the following matrix form

$$
\boldsymbol{H}^{(i)} = B_i (\boldsymbol{Mh})(\boldsymbol{Mh})^\top. \tag{29}
$$

Taking expectation from both sides, we obtain

$$
\mathbb{E}_{\boldsymbol{h} \sim (\boldsymbol{\mu}, \boldsymbol{\Sigma})}\left[ \boldsymbol{H}^{(i)} \right] = B_i \mathbb{E}_{\boldsymbol{h} \sim (\boldsymbol{\mu}, \boldsymbol{\Sigma})}\left[ \boldsymbol{M}(\boldsymbol{hh}^\top)\boldsymbol{M} \right] = B_i \boldsymbol{M}(\boldsymbol{\Sigma})\boldsymbol{M}, \tag{30}
$$

where we assume $\boldsymbol{h}$ is centered. Now recall that $\boldsymbol{M} = \text{diag}(m_1, \ldots, m_d)$ where for each $i$, $m_i \sim 1/\sqrt{\Sigma_{ii}}$ holds. Assume that the covariance matrix $\boldsymbol{\Sigma}$ is symmetric positive definite. Then, it is diagonalizable as $\boldsymbol{\Sigma} = \boldsymbol{U}\boldsymbol{\Lambda}\boldsymbol{U}^\top$ with $\boldsymbol{\Lambda} = \text{diag}(\lambda_1, \ldots, \lambda_d)$, a diagonal matrix with eigenvalues of $\boldsymbol{\Sigma}$. With the transformation $\boldsymbol{M}$, we get

$$
\boldsymbol{M}\boldsymbol{\Sigma}\boldsymbol{M} = \boldsymbol{M}\boldsymbol{U}\boldsymbol{\Lambda}\boldsymbol{U}^\top\boldsymbol{M} = \boldsymbol{U}\boldsymbol{M}\boldsymbol{\Lambda}\boldsymbol{M}\boldsymbol{U}^\top \tag{31}
$$

$$
= \boldsymbol{U} \begin{bmatrix} m_1^2 \lambda_1 & & \\ & \ddots & \\ & & m_d^2 \lambda_d \end{bmatrix} \boldsymbol{U}^\top. \tag{32}
$$

Since the eigenvalues capture the variances along the principal components of the covariance matrix, $m_i^2$, as a reciprocal of a measure of dimension-wise dispersion, is reasonably correlated with $1/\lambda_i$, as demonstrated by Lemma 4, implying $\lambda_j \leq \lambda_i \implies m_j \geq m_i$ with high probability. Therefore, we obtain that

$$
\kappa(\boldsymbol{M}\boldsymbol{\Sigma}\boldsymbol{M}) = \frac{\lambda_{\max}(\boldsymbol{M}\boldsymbol{\Sigma}\boldsymbol{M})}{\lambda_{\min}(\boldsymbol{M}\boldsymbol{\Sigma}\boldsymbol{M})} \approx \frac{m_{\min}^2 \lambda_{\max}(\boldsymbol{\Sigma})}{m_{\max}^2 \lambda_{\min}(\boldsymbol{\Sigma})} \leq \kappa(\boldsymbol{\Sigma}), \tag{33}
$$

which implies the claim. $\qquad \square$

**Lemma 4** (Correlation between dimension-wise varainces and covariance eigenvalues). *Let $\{\boldsymbol{b}_i\}_{i \in d}$ be the set of normalized basis vectors of $\mathbb{R}^d$. Consider a symmetric positive definite covariance matrix $\boldsymbol{\Sigma}$ and its unit eigenvectors $\{\boldsymbol{v}_i\}_{i \in [d]}$. Assume that the eigenvector $\boldsymbol{v}_i$ is a reasonably small perturbation of the basis vector $\boldsymbol{b}_i$ such that $\boldsymbol{v}_i^\top \boldsymbol{b}_i \geq 1 - \epsilon$ for all $i \in [d]$ and a small constant $\epsilon > 0$. Then, for all $i \in [d]$, we have*

$$
|\lambda_i - \Sigma_{ii}| \leq \epsilon \cdot \max_{j \neq i} |\lambda_i - \lambda_j|, \tag{34}
$$

*where $\{\lambda_i\}_{i \in [d]}$ is the set of ordered eigenvalues of $\boldsymbol{\Sigma}$ corresponding to eigenvectors $\{\boldsymbol{v}_i\}_{i \in [d]}$.*

*Proof of Lemma 4.* Note that each diagonal element of the SPD covariance matrix $\boldsymbol{\Sigma}$ can be written as

$$
\Sigma_{ii} = \boldsymbol{b}_i^\top \boldsymbol{\Sigma} \boldsymbol{b}_i = \boldsymbol{b}_i^\top \left( \sum_{j \in [d]} \lambda_j \boldsymbol{v}_j \boldsymbol{v}_j^\top \right) \boldsymbol{b}_i = \sum_{j \in [d]} \lambda_j (\boldsymbol{v}_j^\top \boldsymbol{b}_i)^2. \tag{35}
$$

Table 4: Efficiency Comparison between ACMoE and baseline MoE models

| Model | Compute Speed (ms/it) | Max Memory (K) | #Params (M) |
|---|---|---|---|
| *GLaM* (Du et al., 2022) | 422.62 | 25.69 | 220 |
| GLaM-ACMoE (**Ours**) | 425.15 | 25.72 | 220 |
| *Switch Transformer* (Fedus et al., 2022) | 391.93 | 34.64 | 216 |
| Switch-ACMoE (**Ours**) | 393.29 | 34.68 | 216 |
| *Swin Transformer* (Liu et al., 2021) | 403.36 | 22.00 | 280 |
| Swin-ACMoE (**Ours**) | 408.56 | 22.19 | 280 |

Then, the difference on the left hand side of Eqn. 34 can be bounded as

$$
\left|\lambda_i - \Sigma_{ii}\right| = \left|\lambda_i - \sum_{j\in[d]} \lambda_j (\boldsymbol{v}_j^\top \boldsymbol{b}_i)^2\right| = \left|\lambda_i\left(1 - (\boldsymbol{v}_i \boldsymbol{e}_i)^2\right) - \sum_{j\neq i} \lambda_j (\boldsymbol{v}_j^\top \boldsymbol{b}_i)^2\right|
$$

$$
= \left|\lambda_i \sum_{j\neq i} (\boldsymbol{v}_j^\top \boldsymbol{b}_i)^2 - \sum_{j\neq i} \lambda_j (\boldsymbol{v}_j^\top \boldsymbol{b}_i)^2\right| \tag{36}
$$

$$
= \left|\sum_{j\neq i} (\lambda_i - \lambda_j)(\boldsymbol{v}_j^\top \boldsymbol{b}_i)^2\right|
$$

$$
\leq \max_{j\neq i}|\lambda_i - \lambda_j| \sum_{j\neq i} (\boldsymbol{v}_j^\top \boldsymbol{b}_i)^2
$$

$$
= \max_{j\neq i}|\lambda_i - \lambda_j|\left(1 - (\boldsymbol{v}_i \boldsymbol{b}_i)^2\right) \tag{37}
$$

$$
\leq \epsilon \max_{j\neq i}|\lambda_i - \lambda_j|,
$$

where we used the fact that

$$
\sum_{j\in[d]} (\boldsymbol{v}_j^\top \boldsymbol{b}_i)^2 = \left(\sum_{j=1}^n \left(\boldsymbol{v}_j^\top \boldsymbol{b}_i\right)\boldsymbol{v}_j\right)^\top \left(\sum_{k=1}^n \left(\boldsymbol{v}_k^\top \boldsymbol{b}_i\right)\boldsymbol{v}_k\right) = \boldsymbol{b}^\top \boldsymbol{b} = 1
$$

to obtain Eqn. 36 and Eqn. 37 since the eigenvectors of $\boldsymbol{\Sigma}$ are orthonormal. □

## B IMPLEMENTATION PROCEDURE AND COMPUTATIONAL EFFICIENCY

**Training and Inference.** Given the AC routing scheme requires requires the expert assignment per token from the previous layer, we can only implement AC routing from the second layer on. We incorporate AC routing into both training and inference stages. This is because, firstly, AC routing is designed to offer improvements to both clean and contaminated data, and so even in the presence of completely clean train and test data, it is advantageous to incorporate the AC method into both stages. Secondly, it is commonplace to encounter data contamination only at the test stage and indeed highly possible to encounter it in train as well. Therefore, in the interest of robustness as well, AC routing is incorporated into both stages.

**Computational Efficiency.** Computing the required $\{\boldsymbol{w}_k\}_{k\in[E]}$ for number of experts $E$ requires no learnable parameters and is obtained simply by computing the mean absolute deviation for each set of tokens assigned to the $k^{\text{th}}$ expert. This can be computed using just two computations of the mean – once for the mean per cluster and once again for the mean of the absolute deviations per cluster – done in parallel over all clusters using `torch.index_reduce()` and is of the order $\mathcal{O}(2nd) = \mathcal{O}(n)$ for $n$ tokens. Hence the upper-bound time complexity of the MoE layer is unaffected. We provide in Table 4 additional efficiency analysis in terms of throughput, max GPU memory allocated, and parameters which shows no significant efficiency loss compared to baseline MoE architectures.

# C  EXPERIMENTAL DETAILS AND ADDITIONAL EXPERIMENTS

## C.1  LANGUAGE MODELING

### C.1.1  DATASETS

**WikiText-103.**  The WikiText-103[2] dataset contains around 268K words and its training set consists of about 28K articles with 103M tokens. This corresponds to text blocks of about 3600 words. The validation set and test sets consist of 60 articles with 218K and 246K tokens respectively.

**EnWik-8.**  The EnWik-8 dataset is a byte-level dataset of 100 million bytes derived from Wikipedia that, in addition to English text, also includes markup, special characters, and text in other languages. EnWik-8 contains 90M characters for training, 5M for validation, and 5M for testing.

**Stanford Sentiment Treebank-2.**  The Stanford Sentiment Treebank-2 (SST2) (Socher et al., 2013) is a 2 class corpus with fully labeled parse trees for analysis of the compositional effects of sentiment in language. The dataset consists of 11,855 single sentences extracted from movie reviews. It was parsed with the Stanford parser and includes 215,154 unique phrases from the parse trees, each annotated by 3 human judges.

**Stanford Sentiment Treebank-5.**  Stanford Sentiment Treebank-5 (SST5) (Socher et al., 2013) is a 5 class dataset used for sentiment analysis. It consists of 11,855 single sentences extracted from movie reviews. It includes 215,154 unique phrases from parse trees, each annotated by 3 human judges. Phrases are classified as negative, somewhat negative, neutral, somewhat positive, or positive.

**Banking-77.**  Banking-77 (B77) (Casanueva et al., 2020) is a highly fine-grained 77 class classification dataset comprising 13083 customer service queries labelled with 77 intents.

### C.1.2  MODEL, OPTIMIZER, & TRAIN SPECIFICATION

**Models.**  We use as backbones the Switch Transformer (Fedus et al., 2022) and Generalist Language Model (Du et al., 2022). Table 5 contains the specification over self-attention (SA) layers, feed-forward network (FFN) layers, Mixture-of-Experts (MoE) layers, attention span (Att. Span), embedding size and parameter count for both backbones at small and medium configurations for each pretraining task. All backbones use 16 experts with top-2 expert routing.

Table 5: Language Modeling Backbone Specifications

| Model | SA Layers | FFN Layers | MoE Layers | Att. Span | Embed Size | Params |
|---|---|---|---|---|---|---|
| *WikiText-103 Pretrain* | | | | | | |
| Switch-small | 3 | - | 3 | 256 | 128 | 70M |
| Switch-medium | 6 | - | 6 | 1024 | 352 | 216M |
| GLaM-small | 6 | 3 | 3 | 2048 | 144 | 79M |
| GLaM-medium | 12 | 6 | 6 | 2048 | 352 | 220M |
| *EnWik-8 Pretrain* | | | | | | |
| Switch | 8 | - | 8 | 2048 | 352 | 36M |

**Optimizer.**  All experiments use Adam with a base learning rate of 0.0007. Small configurations use 3000 iterations of learning rate warmup while medium configurations use 4000 iterations.

---

[2] www.salesforce.com/products/einstein/ai-research/the-wikitext-dependency-language-modeling-dataset/

**Pretrain Specification.** For WikiText-103 pretraining, small Switch backbones are trained for 40 epochs with a batch size of 96 and medium Switch backbones are trained for 80 epochs with a batch size of 48. Small GLaM backbones are trained for 60 epochs with a batch size of 48 and medium GLaM backbones are trained for 120 epochs with a batch size of 48. We use 0.01 auxiliary load balancing loss.

For EnWik-8 pretraining, both Switch and GLaM backbones are trained for 80 epochs with batch size 48. We use 0.01 auxiliary load balancing loss.

**Finetune Specification.** For SST2 and SST5 finetuning, we finetune for 5 epochs using Adam and a base learning rate of 0.001 without warmup and a batch size of 16. For B77 we finetune for 50 epochs using Adam and a base elarning rate of 0.00001 without warmup and a batch size of 16.

**Compute Resources.** All models are trained, evaluated, and finetuned on four NVIDIA A100 SXM4 40GB GPUs.

### C.2 IMAGE CLASSIFICATION

#### C.2.1 DATASETS AND ATTACKS

**ImageNet-1K.** We use the full ImageNet dataset that contains 1.28M training images and 50K validation images. The model learns to predict the class of the input image among 1000 categories. We report the top-1 and top-5 accuracy on all experiments.

**ImageNet-A/O/R.** ImageNet-A (Hendrycks et al., 2021b) contains real-world adversarially filtered images that fool current ImageNet classifiers. A 200-class subset of the original ImageNet-1K's 1000 classes is selected so that errors among these 200 classes would be considered egregious, which cover most broad categories spanned by ImageNet-1K.

ImageNet-O (Hendrycks et al., 2021b) contains adversarially filtered examples for ImageNet out-of-distribution detectors. The dataset contains samples from ImageNet-22K but not from ImageNet1K, where samples that are wrongly classified as an ImageNet-1K class with high confidence by a ResNet-50 are selected.

Imagenet-R (Hendrycks et al., 2021a) contains various artistic renditions of object classes from the original ImageNet dataset, which is discouraged by the original ImageNet. ImageNet-R contains 30,000 image renditions for 200 ImageNet classes, where a subset of the ImageNet-1K classes is chosen.

**Adversarial Attacks.** We use produce corrupted ImageNet samples using white box attacks fast gradient sign method (FGSM) (Goodfellow et al., 2014) and projected gradient descent (PGD) (Madry et al., 2017), and black box simultaneous perturbation stochastic approximation (SPSA) (Uesato et al., 2018). FGSM and PGD use a perturbation budget of 1/255 while SPSA uses a perturbation budget 1. All attacks perturb under $l_\infty$ norm. PGD and uses 20 steps with step size of 0.15 and SPSA uses 20 iterations.

#### C.2.2 MODEL, OPTIMIZER, & TRAIN SPECIFICATION

**Models.** Our results are based off of the Swin Transformer (Liu et al., 2021) architecture. This backbone uses 4 base layers of depth 2, 2, 18, and 2. The first two base layers each contain 2 self-attention layers and 2 feed-forward layers. The third base layer contains 18 self-attention layers with alternating feed-forward and MoE layers. The final base layer contains 2 self-attention layers with one feed-forward and one MoE layer. The embedding dimension is 96 and the heads per base layer are 3, 6, 12, and 24. We use 16 total experts and present results for both top-1 and top-2 expert routing. The total parameter count is 280M.

**Optimizer.** We use AdamW with a base learning rate of 1.25e-4, minimum learning rate of 1.25e-7, 0.1 weight decay and cosine scheduling.

**Train Specification.** We train for 60 epochs with a batch size of 128 and 0.1 auxiliary balancing loss.

**Compute Resources.** All models are trained and evaluated on four NVIDIA A100 SXM4 40GB GPUs.

## C.3 ADVERSARIAL ATTACK AT HIGHER PERTURBATION BUDGET

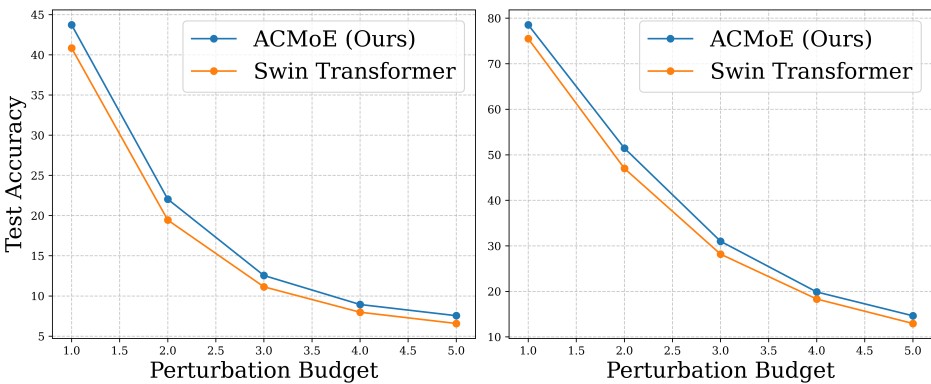

Figure 3: ACMoE and Swin Transformer under PGD attack at increasing perturbation budgets. ACMoE widens its performance gain over Swin at increasingly severe attacks in both top-1 test accuracy (**left**) and top-5 test accuracy (**right**), starting at approximately 7% improvement at 1/255 and ending at just over 10% at 5/255.

Figure 3 shows that for PGD perturbation budgets 1/255 through to 5/255, ACMoE widens its already substantive robust performance gain over Swin, with top-1 and top-5 test accuracy improvements increasing from 7% to approximately 10%.

## C.4 CLUSTER VISUALIZATION

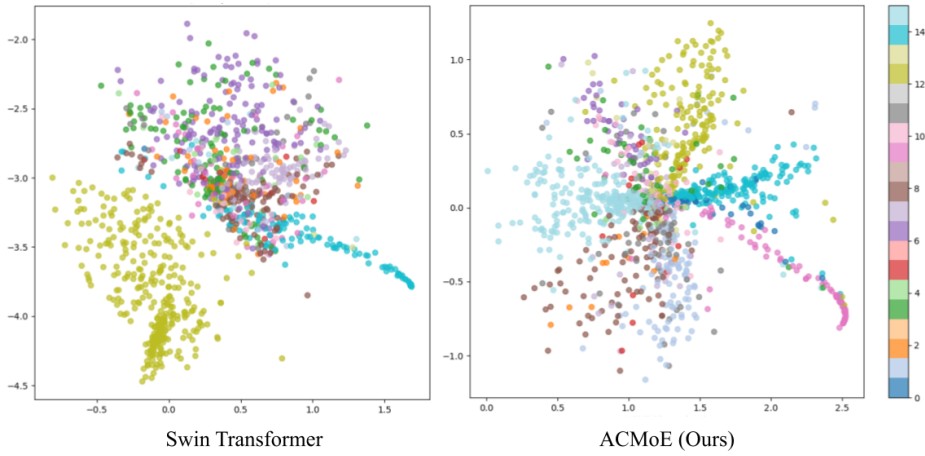

Figure 4: Cluster Visualization on ImageNet. Each token is represented as a point and colored by its assigned expert. **Left**: Swin identifies one cluster clearly (yellow/gold) but otherwise fails to distinguish remaining clusters **Right:** ACMoE learns better-defined expert clusters.

We pass random ImageNet batches through Swin and ACMoE and plot the representations along with their assigned experts, using t-sne to represent the high dimensional data in 2 dimensions. The result is shown in Fig. 4, where we see Swin learns overlapping and indistinguishable expert clusters. ACMoE, on the other hand, performs better in learning the clusters, producing much clearer and better-distinguished clusters.

Table 6: Ablation on Measure of Spread in Switch Transformer (Fedus et al., 2022)

| Measure of Spread | Test PPL ($\downarrow$) |
|---|---|
| Variance | 34.87 |
| MAD | **34.42** |

Table 7: Ablation on Layer Placement in Switch Transformer (Fedus et al., 2022)

| Layer Placement | Test PPL ($\downarrow$) |
|---|---|
| Back Half | 34.95 |
| Alternating | 34.80 |
| Skip 1 | **34.42** |
| Full | 34.88 |

## C.5 ABLATION STUDIES

### C.5.1 MEASURES OF DISPERSION

We present in Tables 6 and 8 results for Switch-ACMoE and Swin-ACMoE when changing the measure of dispersion used in the AC routing transformation (Definition 1) from mean absolute deviation (MAD) to variance. We see mean absolute deviation outperforms variance as a measure of spread. This is an intuitive finding given that squared distances, as used in variance computations, are highly sensitive to outliers. Using mean absolute deviation as an alternative measure of spread reduces this issue and produces a more robust estimate of dispersion. We note that MAD is not the only robust measure of spread. We conjecture that taking interquartile range as an additionally robust measure of spread may produce good results in both clean and contaminated data. We, however, leave this interesting direction to future research as interquartile range poses implementation challenges as it requires designing concurrent linear scans over the expert clusters. MAD, by contrast, requires just two computations of the mean which is easily parallelizable using `torch.index_reduce()`.

### C.5.2 LAYER PLACEMENT

We consider the effect of layer placement in the Switch-medium configuration and in the Swin Transformer (see Sections C.1.2 and C.2.2 for the full model specifications). In particular, Switch is a 6 layer model and Swin is a 24 layer model. With regard to Swin, we focus on the deepest block of depth 18 to implement our ACMoE layers. This is due to the change in embedding size between base layers, meaning we are restricted to this base layer of depth 18. Note further that Swin only uses MoE layers in an alternating pattern with feed-forward networks between each MoE layer. For example, for Switch, a full ACMoE specification would mean placing ACMoE on layers 2,3,4,5,6. For Swin, a full specification means placing ACMoE on layers 4,6,8,10,12,14,16,18. To examine the effect of layer placement we consider the following models:

- *Alternating*: For Switch this means we place ACMoE on layers 2,4,6. For Swin this means we place ACMoE on layers 4,8,12,16.

- *Back Half*: For Switch this means we place ACMoE on just the last 3 layers of the network. For Swin this means we place ACMoE on just the last 5 layers of the network.

- *Skip 2*: For Swin this means we palce ACMoE on layers 8,10,12,14,16,18.

- *Skip 1*: For Switch this means we place ACMoE on layers 3,4,5,6. For Swin this means we place ACMoE on layers 6,8,10,12,14,16,18.

- *Full*: We place ACMoE on every possible layer.

We present in Table 7 results for Switch and Swin ACMoE models when changing the positions of the ACMoE layers throughout the network. The results agree with our expectation that, generally speaking, more ACMoE layers improve performance, but a in some circumstances a threshold is met at the point where ACMoE layers are used too early in the network such that the model has not been able to learn reasonably good approximations of the cluster membership of the tokens yet.

We find that in the Switch backbone, performance improves the more ACMoE layers we add, which agrees with our expectation that more ACMoE layers improve performance. However, we find that top performance is attained when allowing two standard MoE layers to go before the first ACMoE, as opposed to the minimum of 1 standard MoE layer. We conjecture this is because we need to give the model a few layers before the first ACMoE in order to learn decent representations such that we

Table 8: Ablation on Measure of Spread in Swin Transformer

| Measure of Spread | Test Acc. | |
|---|---|---|
| | Top 1 | Top 5 |
| *Swin-Top1* (Liu et al., 2021) | | |
| Variance | 75.06 | 92.49 |
| MAD | **75.39** | **92.56** |
| *Swin-Top2* (Liu et al., 2021) | | |
| Variance | 76.11 | 93.08 |
| MAD | **76.31** | **93.14** |

Table 9: Ablation on Layer Placement in Swin Transformer

| Layer Placement | Test Acc. | |
|---|---|---|
| | Top 1 | Top 5 |
| *Swin-Top1* (Liu et al., 2021) | | |
| Back Half | 75.16 | 92.46 |
| Skip 2 | 75.34 | 92.42 |
| Skip 1 | 75.35 | 92.45 |
| Full | **75.39** | **92.56** |
| *Swin-Top2* (Liu et al., 2021) | | |
| Back Half | 76.16 | 93.02 |
| Skip 2 | 76.10 | 92.93 |
| Skip 1 | 76.29 | 92.98 |
| Full | **76.31** | **93.14** |

have good enough estimated cluster assignments for use in the ACMoE layer. Encouragingly, we find just one additional standard MoE layer is sufficient for the benefits of ACMoE to be obtained.

We find in Table 9 that with Swin, best performance is obtained using ACMoE on every possible layer, again agreeing with our expectation that more ACMoE layers improve performance. With Swin, however, we do not face any drop in performance from placing ACMoE too early in the network, and indeed we see *Full* attaining top performance. We conjecture that Swin does not encounter this issue since Swin uses four layers of feed forward networks before the first MoE layer, and so by the first MoE layer the representations are of reasonably good quality to produce good estimates of the cluster membership.

### C.5.3 RANDOM ABLATION

We show the efficacy of the adaptive clustering transformation $M$ (Definition 1) in our AC router at capturing meaningful feature-wise information by ablating it against an alternate $d \times d$ diagonal matrix made up of normal random variables with mean 1 and standard deviation 0.5 (where we clip any negative values to prevent negative weights). We present in Tables 10 and 11 results for language modeling (using Switch) and image classification (using Swin), which show fairly substantial drops in performance in both backbones. This offers evidence to the claim that our AC routing transformation is meaningfully weighting features to improve routing, and that performance gains of our proposed method do not flow from a kind of implicit regularization of introducing noise into the router.

Table 10: Random Ablation in Switch (Fedus et al., 2022)

| Model | Test PPL ($\downarrow$) |
|---|---|
| *Switch-Random* (Fedus et al., 2022) | 38.17 |
| Switch-ACMoE | **34.42** |

Table 11: Random Ablation in Swin (Liu et al., 2021)

| Model | Top 1 Acc. | Top 5 Acc. |
|---|---|---|
| *Swin-Random* | 74.22 | 91.87 |
| Swin-ACMoE | **76.31** | **93.14** |

### C.6 CLUSTER WEIGHT MIXING

The AC routing scheme estimates the cluster membership of each token based on its highest affinity cluster assigned in the previous layer. We could also further leverage the top-k structure of the MoE models by mixing the cluster-wise feature weights with weights corresponding to the affinities in the top-k routing. For example, if $h$ has affinity scores $\alpha$ and $1 - \alpha$ to clusters $k$ and $k'$ respectively, then we could also obtain the required AC routing transformation for $h$ as $M_{k^*} = \alpha M_k + (1 - \alpha) M_{k'}$. This approach therefore factors in the confidence with which we believe $h$ belongs to cluster $k$ or $k'$, and can be used for integrating ACMoE into higher expert granularity backbones (i.e higher

top-k settings). Tables 12 and 13 show results for computing $M_{k*}$ by mixing the top-affinity cluster weights (Mix 2) in Switch and GLaM with top-2 routing, versus our presented results which compute $M_{k*}$ just based off of the highest affinity cluster (Mix 1). We see that GLaM-ACMoE benefits substantially from cluster weight mixing whereas Switch-ACMoE prefers just using its top affinity cluster weights. For consistency across models, we present in our main body the Mix 1 results, as GLaM-ACMoE already performs extremely strongly using Mix 1 and so we prefer to opt for the added performance gain in the Switch backbone.

Table 12: Results on Cluster Weight Mixing in Switch (Fedus et al., 2022)

| Clusters Mixed | Test PPL (↓) |
|---|---|
| Mix 2 | 34.66 |
| Mix 1 | **34.42** |

Table 13: Results on Cluster Weight Mixing in GLaM (Du et al., 2022)

| Clusters Mixed | Test PPL (↓) |
|---|---|
| Mix 2 | **35.29** |
| Mix 1 | 36.26 |

### C.7 ADAPTIVE CLUSTERING INTEGRATION INTO SOFT MIXTURE OF EXPERTS

We present here results for integrating ACMoE into SoftMoE (Puigcerver et al., 2023). To use ACMoE in the SoftMoe setting, which can be be understood as a top-E routing setting where all experts are active for every token, we compute $M_{k*}$ using cluster weight mixing (Section C.6) over the top-8 highest affinity clusters. We present the performance of Soft-ACMoE on clean data, adversarially attacked data, and ImageNet-A/O/R in the following Tables 14 and 15.

Table 14: Test Accuracy on ImageNet corrupted PGD, FGSM, and SPSA using SoftMoE (Puigcerver et al., 2023) backbone

| Model | Clean Data | | PGD | | FGSM | | SPSA | |
|---|---|---|---|---|---|---|---|---|
| | Top 1 | Top 5 | Top 1 | Top 5 | Top 1 | Top 5 | Top 1 | Top 5 |
| *SoftMoE* (Puigcerver et al., 2023) | 72.86 | 90.92 | 45.29 | 78.91 | 56.95 | 85.60 | 66.59 | 88.70 |
| Soft-ACMoE (**Ours**) | **73.21** | **91.23** | **48.25** | **80.49** | **59.01** | **86.69** | **70.63** | **93.22** |

Table 15: Test Accuracy on Image Classification in Imagenet-A/O/R using SoftMoE (Puigcerver et al., 2023) backbone

| Model | Im-A Top-1 Acc. (↑) | Im-R Top-1 Acc. (↑) | Im-O AUPR (↑) |
|---|---|---|---|
| *SoftMoE* (Puigcerver et al., 2023) | 6.69 | 31.63 | 17.97 |
| Soft-ACMoE (**Ours**) | **6.93** | **32.18** | **18.35** |

We see in Tables 14 and 15 the efficacy of ACMoE in the SoftMoE backbone, offering evidence of the adaptability of our framework into further MoE setups. In particular, the SoftMoE framework models a setting in which expert clusters are highly overlapping, as each token is soft assigned to all experts. Therefore, the performance gains shown in clean and contaminated data of Soft-ACMoE demonstrates that our AC router is well-suited to modeling such a clustering structure.

### C.8 IMAGE CLASSIFICATION IN SWIN TRANSFORMER BASE CONFIGURATION

We further evaluate the performance ACMoE when scaling up model size in Table 16. We integrate ACMoE into the Base configuration of Swin (0.5B parameters) and evaluate on clean ImageNet-1K as well as under adversarial atacks.

### C.9 ROUTER STABILITY

We present in Fig. 5 the routing stability of ACMoE, SMoE, XMoE, and StableMoE in the Switch backbone evaluated on WikiText-103. Routing instability computes over adjacent layers the proportion of tokens that are assigned to different experts across the two layers. Specifically, for $n$ tokens $[h_1, \ldots, h_n]$, we compute at layer $\ell$ the matrix $S^\ell \in \mathbb{R}^{n \times n}$ such that $S_{ij}^\ell = 1$ if the $i^{th}$ and $j^{th}$ tokens are assigned to the same expert in layer $\ell$ and is 0 otherwise. The router instability at layer $\ell$ can

Table 16: Test Accuracy on ImageNet corrupted PGD, FGSM, and SPSA using Swin Base (Liu et al., 2021) backbone

| Model | Clean Data | | PGD | | FGSM | | SPSA | |
|---|---|---|---|---|---|---|---|---|
| | Top 1 | Top 5 | Top 1 | Top 5 | Top 1 | Top 5 | Top 1 | Top 5 |
| *Swin-Base* (Liu et al., 2021) | 79.06 | 94.37 | 44.61 | 79.20 | 59.91 | 87.72 | 68.94 | 89.00 |
| Swin-ACMoE-Base (**Ours**) | **79.25** | **94.42** | **46.28** | **80.24** | **61.78** | 87.55 | **70.18** | **89.33** |

then be calculated as $r^\ell = \mathtt{mean}(|\boldsymbol{S}^{\ell-1} - \boldsymbol{S}^\ell|)$. This metric therefore captures the degree to which tokens that are assigned to the same experts remain together through the model. A high $r^\ell$ indicates the router doesn't maintain consistent expert assignments, as tokens that it considers semantically similar at one layer it considers different at the next.

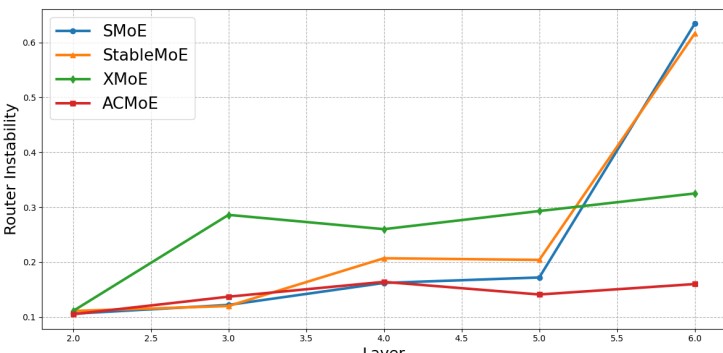

Figure 5: Router Instability of ACMoE, SMoE, XMoE, and StableMoE. ACMoE maintains consistent routing, while baseline routers more frequently change the expert assignments of tokens.

In Fig. 5, we see that baseline routers reach high levels of instability, where in the case of SMoE and StableMoE, at the last layer over 60% of tokens are assigned to a different expert. ACMoE, by contrast, maintains a more consistent, stable assignment through the model, with no more than 20% of tokens changing expert assignment across any layer.

## C.10 DYNAMIC ROUTING

We further test the compatibility of our Adaptive Clustering routing scheme in dynamic top-p routing. In this setting, rather than routing each token to its top-k highest affinity experts in each MoE layer, we route each token to all experts that have affinity over a certain threshold $p$. This setting permits activating more or less experts for different tokens at different layers throughout the model, therefore dynamically assigning experts to tokens. We integrate our AC routing directly into this setting using the same setup as in Section 3, where the AC routing transformation is computed based on the estimated cluster membership of each token using the top affinity assignment of the previous layer. We present the results for Switch transformer on WikiText-103 language modeling in the following Table 17.

For fixed $p$, we set $p = 0.05$. For learnable $p$, we initialize the parameter to 0.05. We select this initialization as it reproduces approximately similar performance in the Switch backbone under default top-2 routing, thereby aiding direct comparison between fixed top-k and dynamic top-$p$ routing. We see in the dynamic routing setting, ACMoE maintains the same consistent improvement over the Switch baseline of roughly 1 full PPL. These results suggest ACMoE is well-suited to the dynamic routing setting.

## D BROADER IMPACT

Our research offers benefits to Mixture-of-Expert (MoE) architectures in both clean and contaminated settings. In particular, our work offers socially beneficial outcomes with regard to defense

Table 17: Results on Top-$p$ Dynamic Routing in Switch Backbone (Fedus et al., 2022)

| Model | Test PPL ($\downarrow$) |
|---|---|
| *Fixed top-k routing* (Shazeer et al., 2017) | |
| *Switch-medium* (Fedus et al., 2022) | 35.48 |
| ACMoE-medium (**Ours**) | **34.42** |
| *Dynamic top-p routing* (Guo et al., 2024) | |
| *Switch-Fixed* $p$ | 35.20 |
| Switch-ACMoE-Fixed $p$ (**Ours**) | **34.14** |
| *Switch-Learnable* $p$ | 34.29 |
| Switch-ACMoE-Learnable $p$ (**Ours**) | **33.49** |

against adversarial attack, which we hope can be used to protect important AI systems from malicious actors. Furthermore, as large language models, many of which are built on MoE backbones, continue to profligate and be used in important societal settings, we hope our improved robustness to data contamination can aid this promising technology to continue to grow and improve in realistic settings of noisy training and evaluation data. Our research also shows substantially faster convergence than comparative baselines. We believe this faster convergence can deliver significant social benefit in terms of reducing the energy requirements of large model training, thereby helping to ease the growing environmental burden of AI training runs. We recognize there will always be risk of misuse with AI systems, however we hope that our work can be used to enhance and protect socially beneficial AI while also decreasing the environmental impact of this technology. We furthermore hope that our research can spur others on to continue building on robust and efficient AI for social good.

