# OpenReview forum: "Tight Clusters Make Specialized Experts"
_ICLR.cc/2025/Workshop/MCDC — MCDC @ ICLR 2025_

### Official Review · Reviewer_ygoQ · 2025-02-27

**Rating:** 7
**Confidence:** 4
**Fit:** 5

**Summary:**

This paper describes an improved token-expert routing method for MoE. It proposes to view the token-expert assignment as a feature-weighted clustering problem and provides a theoretical framework to analyze the robustness and convergence.

**Reason For Giving A Higher Score:**

The theoretical analysis is impressive.

**Reason For Giving A Lower Score:**

The experiments are not so hard to argue.

**Strengths And Weaknesses:**

Pros:

1. The proposed method is supported by rigorous theoretical analyses for its robustness.

2. Experiments on relatively large datasets and both language and vision domains validate the effectiveness of the proposed method compared to established baselines.

Cons:

1. The writing and organization of this paper could be improved. The formular description of the proposed method is unnecessarily long considering that the proposed method is not so complicated. I would recommend saving the space by using straightforward descriptions and moving more experiments in the appendix into the content.

2. The experiments are based on rather small-scale models and MoE methods proposed by 2022. It would be a valuable addition if more experiments were conducted on recent MoE models, especially the language models.

**Suggestions:**

There are some missing references in the appendices, especially A.1.

---

### Official Review · Reviewer_RZE9 · 2025-03-04

**Rating:** 8
**Confidence:** 4
**Fit:** 4

**Summary:**

This paper introduces a novel routing mechanism, the Adaptive Clustering (AC) router, for Mixture-of-Experts (MoE) architectures. The core idea is to determine token-expert assignments within an adaptively transformed space that more effectively uncovers latent data clusters. This transformation is guided by a feature-weighted clustering optimization approach, where features that enhance compact clustering for each expert receive higher weights. The authors present both theoretical and empirical evidence demonstrating the method’s advantages, including faster convergence, improved robustness to data contamination, and superior performance across language modeling and image classification tasks.

**Reason For Giving A Higher Score:**

The paper is well motivated and well written. Provides a solid theoretical analysis and back it up with experiments.

**Reason For Giving A Lower Score:**

N/A

**Strengths And Weaknesses:**

# Strengths

1. The paper addresses a very important topic: how to optimally assign tokens to each expert, which is a crucial contribution.
2. The paper provides solid theoretical results.
3. Experimental results are convincing. Specially Figure 2 is amazing. Faster convergence can translate into massive cost savings in large scale.

# Weaknesses

1. The provided experiments are conducted with small scale models (220M). It is unclear how these insights would scale with larger models.
2. More ablations with hyperparameters would have been helpful. But since this is a workshop paper, that is acceptable.

**Suggestions:**

1. At least a single large scale model experiment would have been more illuminating.
2. It is unclear how this method works with deeper models (more layers) as there can be an error propagation as the model grows in depth. It is good to include a discussion on this.

---

### Official Review · Reviewer_3bPS · 2025-03-04

**Rating:** 7
**Confidence:** 4
**Fit:** 5

**Summary:**

## Summary
The authors address the problem of expert-token "misrouting" within the router component of Sparse Mixture of Experts (SMoE) models/layers with an "Adaptive clustering" (AC) router. in the AC MoE layer, the expert-token score is modified to include a simple and computationally cheap weighting/scaling transformation applied on the token vectors, as inspired by clustering optimization technique of finding feature weighting to represent how much a given cluster "cares" about each dimension/feature proportionally. The authors claim that this simple weighting leads to:
- faster convergence for training, as the Hessian of the loss function has a lower condition number compoared to the loss for the "naive" MoE with a router that doesn't include the weighting transformation
- robustness, as a result of the weighting transform is that the token clusters corresponding to the experts have better separability, thus leading to higher tolerance for noise in the token vectors
- better overall performance, because the clusters are more semantically distinct and corresponding experts specialized

The form the transformation takes is a $ d \times d$ diagonal matrix with entries being the reciprocals of each dimension's "spread" (the authors use mean absolute deviation for "spread") within the token vectors assigned to that expert cluster.

**Reason For Giving A Higher Score:**

Very strong experimental results, compelling theoretical analysis, low cost, evaluated against both clean and "corrupt" test sets

**Reason For Giving A Lower Score:**

assumptions of theoretical analysis not described deeply and experimental set up is not compared to these assumptions

**Strengths And Weaknesses:**

## Strong
The claim that the weighting matrix acts as an effective preconditioner on the Hessian with respect to the expert vector $e_i$ is straightforward and convincing. paired with empirical results in Figure 2 and the fact that the method requires relatively little additional computation, it is compelling.

Performance improvements are modest but consistent against appropriate baselines using appropriate evals, which paints a convincing picture.

Empirical results for robustness support that the AC router has significant effect in handling noise and adversarial inputs.

The fact that the scaling matrix can approach an identity matrix for uniform values of spread across dimensions frame this method as something that can be implemented "just in case" with negligble downside due to the low resource requirement.


## Weak

The cluster separability boost shown in lemma 2 is based on data from a gaussian mixture model with the same number of sources as the number of clusters. A less idealized case would strengthen the conclusions of the authors

The derivation of the weights is based on tokens belonging to a single cluster. The cluster mapping function r, which acts as an analogy to the MoE router, is described as a classifier, which would mean our router sends a token to only one expert. Since the paper concerns the more general case of sending tokens to multiple experts, why can we expect these weights derived from single-cluster assignment to be useful. The vision model the authors trained does indeed use a top 1 router, but the text model uses top 2 routing

The formulation of AcMoE layers in definition 2 uses the clustering/asignment from the previous layer as the scaling matrix for the current layer, which implies that the clustering of the current layer should be similar to the clustering of the previous layer. Why is that assumption not an issue?

### Questions

The derivation of this scaling matrix is shown as coming from a clustering optimization, in which optimal weights are found analytically for a fixed clustering scheme. The solution for the weights in this context are shown in equation 5 and the proof backing it is shown in the appendix. However, there is a leap from eqn 5:
$$
w_{qk} = \frac{\lambda/d}{s_{qk} + \alpha_k}
$$
to the authors matter of factly describe the weights as proportional to the reciprocal of the spread
$$
w_{qk} \propto \frac{1}{s_{qk}}
$$
without explanation for why we can effectively ignore the $\alpha_k$ constant term. In the proof of lemma 3 in the appendix A1, it is shown based on the shape of the $\phi$ functions functions, there are $d$ values of $\alpha_k$ based on the root finding formulation and that the roots will lie in the intervals $ (s_{q-1,k}, s_{qk}) $ except for the rightmost interval which is $ (s_{d,k}, \infty ) $.
So all put together, I am not sure why the alpha terms in the denominator can be ignored. Is it because you can say that it is similar to one of the $s_qk$ and thus we can combine s and alpha and count it as a multiplicative constant?

**Suggestions:**

## Suggestions
- Highlight assumptions in theoretical analysis more explicitly and address how these assumptions are not met in experimental data/setup.

- Address the load balancing aspect of routing. Does the AC router still allow for experts to be utilized well?

- Cover more training convergence results

### Nitpicks
- a few times in appendix A1 text refers to "Eqn. ??" which I assume is supposed to be eqn 5

---

### Official Review · Reviewer_gZGU · 2025-03-05

**Rating:** 7
**Confidence:** 4
**Fit:** 5

**Summary:**

The paper presents an interpretration of routing in MoE models as a form of clustering. From this interpretation, the Adaptive Clustering router is introduced, which computes token-expert assignments in a transformed space. The experiments show that an MoE with this router achieves better results than Switch Transformers and GLaM for language modeling, and better than Swin Transformers for image classification.

**Reason For Giving A Higher Score:**

In order to give a higher score, I would need to see a convincing explanation on the causality issues that I raised, and a proper comparison with other related works.

**Reason For Giving A Lower Score:**

Even if the language model results are invalid due to causality breakeage, studying the router through the lens of clustering is also interesting. Plus, the method is perfectly applicable for encoder models (e.g. image classification). Thus, I could maybe downgrade the rating a bit due to the causality concerns, but I still find the paper very interesting.

**Strengths And Weaknesses:**

**Strengths**
- The proposed approach is well fundamented, starting from a (not novel but) refreshing perspective.
- All the theoretical guarantees and propositions are well explained and proved with rigor, with quite reasonable assumptions (in most cases, see later).
- The proposed router is evaluated on different Transformer-based architectures and tasks (language modeling and image classification).
- The paper is excellently written, kudos to the authors.

**Weaknesses**
- The matrix $\mathbf{M}_{k^*}^{l - 1}$ is based on the token assignments (from the previous layer). This can clearly break causality in auto-regressive models (the weight of a token at time $t$ may depend on the clustering of future tokens), so it's not clear how the authors (correctly) applied this method for language modeling. This means that the PPL and accuracy evaluations could be potentially invalid.
- Comparison with other baselines is lacking. For instance, "On the Representation Collapse of Sparse Mixture of Experts" (Chi et al., 2022) also suggests using a (low-rank) linear projection to compute the router weights. "ModuleFormer: Modularity Emerges from Mixture-of-Experts" (Shen et al., 2023) uses an MLP instead of a linear projection. Finally, clustering and Optimal Transport can be related, and theres a plethora of works using different OT-related approaches for MoEs (e.g. see the survey "Routers in Vision Mixture of Experts: An Empirical Study").
- Many of the theoretical claims are made based on fixed tokens and routing parameters. However, these are not fixed when we train the model. Thus, claims regarding the "optimality" of AC may be irrelevant in practice (i.e. Proposition 1 and 2).

**Suggestions:**

Address the mentioned weaknesses. In terms of writing suggestions, I find the paper of excellent quality.

---

### Decision · Program_Chairs · 2025-03-06

**Decision:**

Accept

**Comment:**

This work proposes a new router for Sparse MoE enabling more specialized experts. This is relevant to the workshop, as it improves current modular capabilities. All the reviewers all recommend acceptance, and we're please to accept it to the workshop.